# Malaria Vaccines: From the Past towards the mRNA Vaccine Era

**DOI:** 10.3390/vaccines11091452

**Published:** 2023-09-04

**Authors:** Maria E. Tsoumani, Chrysa Voyiatzaki, Antonia Efstathiou

**Affiliations:** 1Department of Biomedical Sciences, University of West Attica, 12243 Aigaleo, Greece; mtsoumani@uniwa.gr (M.E.T.); cvoyiatz@uniwa.gr (C.V.); 2Immunology of Infection Group, Department of Microbiology, Hellenic Pasteur Institute, 11521 Athens, Greece

**Keywords:** malaria, vaccines, mRNA, peptide vaccines, protein vaccines, *Plasmodium*, *P. falciparum*, DNA vaccines

## Abstract

*Plasmodium* spp. is the etiological agent of malaria, a life-threatening parasitic disease transmitted by infected mosquitoes. Malaria remains a major global health challenge, particularly in endemic regions. Over the years, various vaccine candidates targeting different stages of *Plasmodium* parasite life-cycle have been explored, including subunit vaccines, vectored vaccines, and whole organism vaccines with Mosquirix, a vaccine based on a recombinant protein, as the only currently approved vaccine for *Plasmodium falciparum* malaria. Despite the aforementioned notable progress, challenges such as antigenic diversity, limited efficacy, resistant parasites escaping protective immunity and the need for multiple doses have hindered the development of a highly efficacious malaria vaccine. The recent success of mRNA-based vaccines against SARS-CoV-2 has sparked renewed interest in mRNA vaccine platforms. The unique mRNA vaccine features, including their potential for rapid development, scalability, and flexibility in antigen design, make them a promising avenue for malaria vaccine development. This review provides an overview of the malaria vaccines’ evolution from the past towards the mRNA vaccine era and highlights their advantages in overcoming the limitations of previous malaria vaccine candidates.

## 1. Introduction

Malaria, a life-threatening mosquito-borne infectious disease caused by the *Plasmodium* parasite, continues to pose a significant global health burden as half of the world’s population lives at risk of infection, particularly in tropical and subtropical regions. Based on the latest World Health Organization (WHO) data, malaria cases increased up to 247 million, with reported 619,000 deaths in 2021, as during the COVID-19 pandemic, the prevention and diagnosis efforts were disordered [1]. Five different parasite species can cause malaria in humans, namely *Plasmodium falciparum, P. vivax, P. ovale, P. malariae*, as well as *P. knowlesi,* which infects non-human primates in Southeast Asia and can also infect humans (“zoonotic” malaria) [2]. Of these species, *P. falciparum* infection commonly results in severe malaria cases, which require prompt treatment to avoid death. In fact, *P. falciparum* was documented as the etiological agent of 95% of malaria cases and 96% of malaria deaths in 2021, and unfortunately, 80% of all malaria deaths were accounted for in children under 5 years old in the African continent [1].

Malaria can be a devastating disease, but there are continuous efforts to prevent illness and death by integrating control strategies that combine vector control measures, improved diagnostics, and access to prompt and effective treatment. Despite the progress in malaria mitigation measures, the intricate life cycle of the *Plasmodium* parasite, involving both humans and female *Anopheles* mosquitoes as its hosts, contributes to the complexity of malaria transmission and its ability to persist in endemic areas [3]. In the mammalian host, *Plasmodium* parasites differentiate in different forms, starting with sporozoite form inoculated during a blood meal of an infected female *Anopheles* mosquito into the human host skin. Subsequently, the sporozoites invade the hepatocytes in the liver, where they mature, replicate, and release merozoite forms, which eventually invade erythrocytes, where they transform from rings and trophozoites to the schizonts stage. The ruptured schizonts release merozoites that can infect fresh blood cells [4,5].

The Global Technical Strategy for Malaria (2016–2030) has a set target to reduce 90% of malaria incidents and mortality rates around the world; thus, there is a need for novel, efficient preventive antimalarial strategies [6]. Although vaccination represents a crucial pillar in the comprehensive approach to malaria control, the development of an effective malaria vaccine has been difficult, primarily due to the complex biology of the parasite and the intricate host immune response required for protection [3]. However, recent advancements in vaccine research and the identification of promising vaccine candidates have renewed hopes for developing effective vaccines for malaria.

Our goal is to provide a comprehensive overview of the current landscape and progress in malaria vaccine development. The current review aims to discuss the challenges posed by malaria and new strategies that can be used to develop effective malaria vaccines. Hence, we will delve into key aspects of vaccine development through the years and the progress from the traditional vaccine development methods towards the mRNA vaccine era in the fight against malaria.

## 2. Vaccines That Target the Sporozoite Form of *Plasmodium* Parasite

Until now, different vaccine strategies have been developed targeting different stages of *Plasmodium* species. Vaccines that target the sporozoite stage aim to prevent infection of the liver and induce immunity against the parasites [7]. At first, whole-killed sporozoites were tested as a malaria vaccine in animal studies, offering only partial protection [8].

Significant progress in the development of malaria vaccines was reported in 1967 by vaccinating mice using irradiated sporozoites (metabolically active and motile but non-replicating) capable of inducing sterile immunity (no detectable parasitemia) to the disease. The radiation-attenuated sporozoite approach was rapidly implemented in humans in many dose and regimen optimization studies [9]. *P. falciparum* (*Pf*) sporozoite vaccine (*Pf*SPZ) from the NF54 isolate manufactured by Sanaria Inc. (Rockville, MD, USA) met all the regulatory requirements needed for testing in humans (Table 1). Studies in animals revealed that *Pf*SPZ should be administered only intravenously (IV) in order to confer protection. Later, for a quick and more precise evaluation of the vaccine’s efficacy, the controlled human malaria infection (CHMI) model was developed. CHMI is considered a successful model since fewer subjects are required in clinical trials [7]. Consequently, studies in malaria-naïve volunteers in the United States who received four doses of the *Pf*SPZ vaccine provided evidence for long-term protection (up to 14 months) by inducing sterile immunity against homologous CHMI [10,11].

Subsequent studies aimed to evaluate if the vaccine’s efficacy persists in different age groups, in individuals previously exposed to the parasite, in endemic areas and the durability of protective efficacy. Studies in African countries have shown that the direct venous inoculation (DVI) route of administration can prolong the vaccine’s protection in adults [12]. DVI route induces circulating *Pf*CSP-specific antibodies as well as circulating and liver-resident T cell responses [13] (Table 1). Oneko et al. have shown that there are differences in the vaccine’s efficacy between infants, children and adults, reporting how cellular immune response is affected by age. Indeed, in this comparative study between age groups, it was reported that although infants vaccinated with the highest dose of *Pf*SPZ generated the highest levels of antibodies, the *Pf*SPZ-specific T-cell responses, indicative of the induction of protection, were not detected. The lack of γδ T cells may explain the vaccine’s modest efficacy observed at three months instead of six months, which was the primary endpoint [13]. Furthermore, individuals who were previously exposed to *P. falciparum* exhibited low vaccine efficacy. We could hypothesize that pre-existing natural immunological responses interfere with the development of robust immune responses following vaccination with *Pf*SPZ [14]. Moreover, despite parasite diversity in endemic areas, clinical trials conducted in countries such as Tanzania and Balonghin, Burkina Faso, have provided encouraging results since they have shown that an optimized *Pf*SPZ vaccine dosage is well tolerated, safe and effective in African adults throughout an entire malaria season [15,16,17].

Another vaccine approach combines replication–intact sporozoites with antimalarial drug prophylaxis, such as chloroquine and mefloquine, to prevent malarial illness [18,19]. *Pf*SPZ-CVac manufactured by Sanaria Inc. (CVac = Chemoprophylaxis Vaccine) can induce sterile immunity in malaria-naive volunteers, whereas it cannot confer significant protective efficacy in malaria-exposed adults [20]. *Pf*SPZ-CVac is a stronger immunoantigen compared to *Pf*SPZ since the infectious *Pf*SPZ in *Pf*SPZ-CVac expresses ∼4500 different proteins, including blood-stage proteins, whereas *Pf*SPZ expresses ~1000 proteins. Thus, there are dramatically more parasites and antigens presented to the immune system per *Pf*SPZ injected with *Pf*SPZ-CVac [21]. Nevertheless, the dose schedules of this vaccine are not standardized. Compressing the*Pf*SPZ–CVac regimen to 28 and 10 days proved safe and simultaneously maintainedhigh efficacy (67 and 63%, respectively). However, the 10-day regimen induced more robust cellular and humoral immune responses (Table 1). This could be attributed to the continuous exposure to liver-stage parasites, which in nature lasts about 6.5 days. The duration of liver-stage exposure is a factor in optimizing *Pf*SPZ–CVac immunogenicity [22]. In conclusion, *Pf*SPZ-CVac could be used in elimination campaigns in endemic areas where the population is already exposed to natural malaria transmission [21].

Alternative strategies for *P. falciparum* attenuation are the genetic modifications procedure that incorporate mutations to the parasite that lead to the arrest of parasite development at various points during liver infection in humans. Identification of genes that are upregulated in infective sporozoites resulted in the development of a *P. falciparum* early liver stage-arresting triple knockout parasite (p36‾/p52‾/sap1‾), namely *Pf*GAP3KO. *Pf*GAP3KO has been assessed for its immunogenicity in humans, showing potent sporozoite infection-blocking antibodies, but no data are available for T-cell responses [23,24]. In a CHMI study, *Pf*GAP3KO was safe, immunogenic and capable of achieving protection in half of tested malaria-naïve study participants [25]. Another *P. falciparum* genetically attenuated parasite was created by the deletion of two genes loci, slarp and b9, each governing independent and critical processes for successful liver-stage development. *Pf* double-knockout (*Pf*Δb9Δslarp) SPZ was manufactured by Sanaria (*Pf*SPZ-GA1 vaccine), and a clinical trial was conducted to assess its safety, immunogenicity and efficacy in malaria-naïve volunteers in comparison with *Pf*SPZ group. The results showed that the vaccine was safe and well-tolerated when administered by DVI. Moreover, it induced cellular and humoral immune responses. Regarding efficacy, the authors observed an unexpectedly low efficacy in the *Pf*SPZ reference group, which limited the ability of the authors to interpret the observed data for the *Pf*SPZ-GA1 vaccine [26].

## 3. Circumsporozoite Protein Subunit Vaccines

The circumsporozoite protein (CSP) on the surface of the malaria sporozoite is represented early in the liver phase of infection, and it is considered a major antigen component [27]. The RTS,S/AS01_Ε_ (RTS,S), the most advanced of the subunit malaria vaccines, is a pre-erythrocytic *P. falciparum* vaccine that consists of a protein (RTS) of the NANP repeat and C-terminal portions (R and T, respectively) of the NF54 strain of *P. falciparum* CSP, fused with the hepatitis B virus surface antigen (HBsAg; the S portion). It is administered with a liposome-based adjuvant (AS01), which is used to enhance the immune response to vaccination through antibodies and CD4^+^ T cells [7,28] (Table 1, Figure 1). The antibodies after vaccination have to act quickly in order to prevent sporozoites’ invasion of hepatocytes (naturally, sporozoites reach the liver within 30 min after a mosquito bite) and also to elicit a cellular response, enabling the destruction of infected hepatocytes [14,29]. Phase 3 trials showed a median vaccine efficacy against malaria of 55.1% (95% confidence interval [CI], 50.5–59.3%) over 12 months after vaccination when delivered according to a 0-, 1-, and 2-month schedule in children aged 5–17 months at first vaccination [30]. Despite the limited efficacy and the fact that it wanes over multiple years, RTS,S /AS01_E_ is the first malaria vaccine (trade name Mosquirix, GlaxoSmithKline Biologicals SA) that gained WHO approval in October 2021 [1] (Table 1). It has been approved for vaccination in children under the age of two residing in regions of moderate to high malaria transmission. Moreover, it is recommended to be provided in countries with seasonal malaria transmission. Implementation of Mosquirix vaccination has reduced hospital admissions for severe malaria by around 30% (modest efficacy) [31]. Moreover, a booster vaccination schedule, including a delayed and fractional third dose showed 87% efficacy in malaria-naïve adults but not in children [32]. The modest efficacy of RTS,S/AS01_E_ predicates the development of an optimal vaccine against malaria a clinical need.

R21 vaccine is the next-generation pre-erythrocytic *P. falciparum* vaccine with the aim to be more immunogenic than the RTS,S/ASO1_E_. R21 particle is different from the RTS,S since it contains only the CSP-HBsAg fusion protein, resulting in a higher density of the CSP on the surface [33] (Figure 1). It has been developed at the University of Oxford and is currently manufactured by the Serum Institute of India (Pune, India). As an adjuvant, Matrix-M™ (MM) has been used, a formulation that is similar to a COVID-19 vaccine, which consists of the SARS-CoV-2 spike (S) glycoprotein (NVX-CoV2373) and-MM adjuvant (Novamax, Gaithersburg, MD, USA) [34] (Table 1). R21/MM vaccine has reached the WHO’s goal of at least 75% efficacy over 12 months in African children under the age of two who had been previously exposed to malaria [35]. One limitation of this study is that it had been conducted prior to the peak malaria season, as antibody levels had declined markedly at 12 months post-vaccination. The funding organization extended the clinical trial in order to determine vaccine efficacy over 2 years across different malaria transmission settings (NCT04704830) [7,14]. At the same target group, a phase 1/2b clinical trial was conducted in order to evaluate if a booster dose of R21/Matrix-M at 12 months after the initial three doses maintains the observed immunity. Authors reported that the booster dose exerts high efficacy (over 75%) against first and multiple episodes of clinical malaria as well as high antibody titers [36]. With regard to the safety profile of the vaccine, no major adverse effects have been reported with R21/MM immunization [35,36]. Due to its high efficacy and favorable safety profile, the R21/MM vaccine has been licensed for use in three African countries: Ghana, Nigeria and Burkina Faso.

## 4. Viral-Vector-Vaccines

Viral-vectored vaccines (Figure 2A) were designed in the pursuit of the aim to exceed the protection rate of the RTS,S vaccine by enhancing the cellular immunity against the liver stage of *P. falciparum* [37]. Despite the fact that several viral-vector vaccines did not reach this goal, in the last several years, the progress of this approach has been accelerated considerably. Currently, one of the most advanced viral-vector vaccines is chimpanzee adenovirus 63 (ChAd63) and modified Vaccinia Ankara (MVA) (Figure 2B). ChAd63 and MVA encode the thrombospondin-related adhesion protein pre-erythrocytic antigen and the multiple epitope string (ME-TRAP) in order to prime an immune response. This particular prime-boost immunization approach (MVA is used as a prime booster) induces higher CD8^+^ T cell responses than single vector immunization, and it confers 21% sterile short-term protection as determined by CHMI in malaria-naive adults [28,34]. Moreover, its efficacy has been verified in a field study since vaccination reduced the risk of infection by 67% among adults living in a malaria-endemic area in Kenya [38]. As described above, the development of all malaria vaccine candidates targets one stage of the complex life of *P. falciparum*. Recently, a multi-stage vaccine regimen, namely human adenovirus 5 (AdHu5)- adeno-associated virus serotype 1 (AAV1) prime-boost (Figure 2B), was generated, and in animal studies, it provided significant efficacy as well as induced high-titer antibody responses. It expresses either *P. falciparum* pre-erythrocytic PfCSP or sexual stage *Pf*s25 antigen [39].

## 5. Erythrocytic Vaccines (Blood-Stage Vaccines)

Erythrocytic vaccines are a unique category of malaria vaccines since their mechanism of action is to block the invasion of red blood cells by the merozoites (after the completion of the pre-erythrocytic stage) [40]. They target antigens highly expressed on the surface of merozoites, namely erythrocyte-binding antigen-175 (EBA-175), apical membrane antigen-1 (AMA-1), glutamate-rich protein (GLURP), serine repeat antigen 5 (SERA5) and merozoite surface proteins (MSPs). However, this strategy has provided disappointing results from the clinical studies, maybe because the induced antibodies cannot act as quickly as required in order to trap the numerous merozoites outside the erythrocytes [27]. Moreover, the above antigens are highly polymorphic. Recently, the recombinant SE36 antigen formulated with aluminum hydroxyl gel (BK-SE36) has been assessed for its immunogenicity and safety in phase I [41] and phase Ib [42] trials. BK-SE36 malaria vaccine was safe and well-tolerated as well as highly immunogenic when given to healthy semi-immune children under 5 years old [42]. Phase II clinical trial has to be designed in order to verify its effectiveness.

## 6. Nucleic Acids in Malaria Vaccine Development

As previously mentioned, the Mosquirix vaccine has shown significant success in reducing severe malaria cases and lowering child mortality, and currently, more than one million children living in areas with moderate-to-high malaria transmission have received the vaccine, marking its widespread use [1]. However, Mosquirix has encountered certain limitations, including lower efficacy in specific age groups besides children, lack of durable immune responses and the necessity of three to four booster doses to achieve reasonable efficacy [29,43]. Consequently, the search for a more robust and effective malaria vaccine is still ongoing.

The high demand for implementation of fast-track stages in vaccine research and development introduced early the idea of considering nucleic acids as a tool for the evolution of third-generation vaccines [44]. Using genetic technology to develop DNA or RNA vaccines has been introduced 30 years ago in basic research, and the advantages of such technology have been outlined compared to conventional methods. The approach of nucleic acid vaccine development consists of isolating pathogens in the field, gene sequence and in silico recognition of antigens, evaluation of vaccine efficacy and efficiency, production and clinical trials [45]. Improvements in genetic engineering, bioinformatics and computational approaches provide an immense boost in the rapid antigen design [45,46]. However, at first, the production of mRNA vaccines against infectious diseases faced significant controversy due to concerns surrounding RNA instability, challenges in large-scale manufacturing, vaccine reliability, and potential implications [44,45]. Hence, the first attempts at third-generation vaccine development focused on DNA.

### 6.1. DNA Vaccines

To address the challenges associated with malaria treatment and vaccination, such as drug-resistant malaria and the complex cell cycle of the malaria parasite, the versatile DNA vaccine technology has gained attention. This technology enables the convenient production of vaccines capable of targeting multiple antigens from both the preerythrocytic and erythrocytic stages of the malaria parasite [5]. Simultaneously, DNA-based vaccines are being recognized as highly promising due to their straightforward production, cost-effectiveness, extended shelf-life, independence from a cold chain, and capacity to stimulate both humoral and cellular immune responses [5].

Several *Plasmodium* proteins have been in the spotlight for exploitation in the form of DNA vaccines in preclinical trials. The most potent one, CSP, has been widely examined by different research groups in order to optimize the appropriate codon modifications, gene sequence tag additions, regime alterations and route of administration that could stimulate the desired immune response and protective immunity [47,48,49,50]. Nevertheless, the process of codon optimization did not yield the expected strong CD4+ and CD8+ T cell responses, suggesting that the impact of mammalian codon optimization may vary depending on the antigen. In the context of vaccines designed to induce T cell-dependent protective immunity in this malaria model, it did not appear to confer any advantages [48]. Consequently, immunization of experimental models of mice in different studies with *Plasmodium* CSP resulted in in vivo antibody production and decreased the parasitic levels in the liver depending on the method of administration and the construct [51,52].

The induction of a specific immune response is essential for establishing protective immunity against *Plasmodium* parasites. Particularly, CD8^+^ T cells have been identified as the primary effectors in the fight against malaria. Therefore, the initial hypothesis that scientists sought to test was whether DNA vaccines could elicit this type of immune response. Wang et al. immunized primates (rhesus monkeys) with four plasmid DNAs expressing pre-erythrocytic (sporozoite/liver) stage proteins. These proteins individually have been previously shown to be immunogenic in mice, namely the *Pf*CSP, *Pf* sporozoite surface protein 2, *Pf* protein exported protein 1, and *Pf* liver-stage antigen 1. Upon immunization, the monkeys exhibited antigen-specific cytotoxic T lymphocytes, which were attributed solely to CD8^+^ T cells [53], proving that this strategy is highly promising to be implemented in humans.

Numerous other antigenic proteins have been identified as antigenic and potent vaccine candidates, either in vitro or in silico, and their corresponding DNA vaccine candidates have been evaluated in vivo (i.e., *Pf* sexual stage surface antigen s25, GPI8p transamidase-related protein, merozoite surface protein-1). However, the assessment of these DNA vaccines has been limited to preclinical studies conducted in murine experimental models [5,54,55].

### 6.2. Malaria Vaccines in mRNA Era

The success of COVID-19 mRNA vaccines has paved the way for companies to utilize this technology in the development of RNA-based vaccines for various infectious diseases, including malaria. The obstacles associated with mRNA vaccines, which previously arose from the complexities of advancements in RNA biology and chemistry, are currently being successfully addressed. This progress is facilitating the broad adoption of this technology. The breakthroughs in self-amplifying mRNA vaccines and the development of lipid-based formulations (Figure 3A) represent a disruptive innovation, introducing a novel approach to vaccine production. In contrast to the existing mRNA vaccines produced for COVID-19, which introduce a limited amount of non-replicating mRNA, self-amplifying RNA vaccine technology includes the necessary replication machinery (self-amplifying RNA vectors containing the nonstructural protein genes that encode a viral replicase, 5′ and 3′ sequences important for replication, and a subgenomic promoter derived from alphavirus vectors) [56] (Figure 3B). This enables intracellular RNA amplification and consequently results in abundant protein expression. The novel mRNA technology approach utilizes a straightforward, synthetic, rapid, and cell-free process, potentially enabling the creation of numerous advanced products in the future. These advancements hold promise to surpass the technology employed in mRNA COVID-19 vaccines [45,57,58].

Thus, leveraging the recent achievements in vaccine development for COVID-19, RNA technology is now being utilized to develop hopeful antimalarial candidate vaccines. For instance, the self-amplifying RNA technology has been used for the development of a vaccine that targets the antigen *Plasmodium* macrophage migration inhibitory factor (PMIF), which is secreted by the parasite and serves to suppress the host’s inflammatory response to the infection, particularly targeting the T-cell response [59]. Immunization of mice resulted in the stimulation of enhanced differentiation of memory CD4 T-cells and liver resident CD8 T-cells, as well as an elevation in antibody levels targeting the parasite. Ultimately, this leads to protection against reinfection and provides defense against both liver and blood-stage infections [60]. With these promising outcomes, there is a possibility that human clinical trials will be conducted in the near future to assess preliminary safety and immunogenicity. Very recently, a self-amplifying mRNA vaccine packaged in cationic liposomes has been developed targeting the blood-stage antigen reticulocyte binding protein homologue 5 (*Pf*RH5) of *P. falciparum.* Mice immunized with the vaccine elicited antibodies that recognized the native protein expressed in *P. falciparum* schizont extracts and inhibited the growth of the parasite in vitro [56]. Mallory et al. used the *Pf*CSP in order to incorporate its mRNA in lipid nanoparticles before being administrated to mice [59]. Due to the insufficient production of protective antibodies, following immunization, it was determined that a booster immunization was necessary to induce a robust immune response. As a result, mice that were vaccinated with mRNA-lipid nanoparticle (LNP) demonstrated partial protection against the disease and exhibited improved survival rates following parasitic infection, particularly after a 6-week immunization schedule [59]. The *Pf*CSP mRNA-LNP vaccine exhibits the ability to disrupt malaria infection in mice at a stage prior to the parasite entering the red blood cells, resulting in complete protection against the disease. This remarkable outcome positions it as a promising candidate for a pre-erythrocytic malaria vaccine.

Based on the same hypothesis, Hayashi et al. conducted an experiment using murine models, where mRNA- LNP vaccines containing antigens *Pf*s25 and *Pf*CSP were administered either individually or in combination. The immune responses generated by the mRNA-LNP vaccines surpassed those induced by the corresponding DNA vaccine formulation immunization [61]. *Pf*s25 mRNA-LNPs induced antibody responses that effectively prevented malaria transmission to mosquitoes, and four immunizations with *Pf*CSP mRNA-LNP protected mice against sporozoite challenge [61]. The current endeavor advocates for a combined approach, utilizing vaccines that target both the infectious stage and the sexual/midgut stages of malaria. This approach is anticipated to play a crucial role in disrupting malaria transmission, which is essential for achieving the goal of elimination [61].

The highly secreted and conserved protein amongst *Plasmodium* species, Cell-Traversal protein for Ookinetes and Sporozoites (CelTOS), has been identified as a potential protective antigen [62,63]. Consequently, various research groups have assessed the effectiveness of CelTOS vaccines and demonstrated their ability to trigger robust immune responses, both humoral and cellular, that can confer functional immunity [4,64,65,66]. For instance, Waghela et al., upon optimization of the mRNA modifications, performed a three-dose regimen of CelTOS mRNA-LNP immunization in mice, which induced antigen-specific cellular cytokine responses accompanied with a sufficient capability to mount *Pf*CelTOS-specific antibody responses [4]. Consequently, the aforementioned efforts emphasized the importance of optimizing antigen modifications to achieve the most effective design for the development of a malaria vaccine.

The mosquito saliva protein AgTRIO has also been used for the generation of an mRNA-LNP vaccine. Upon immunization of mice with AgTRIO mRNA-LNP, a robust humoral response was evoked, including AgTRIO IgG2a isotype antibodies. Moreover, immunized mice had reduced hepatic parasitic load after *Plasmodium berghei*-infection and thus manifested increased survival [67]. A mosquito AgTRIO mRNA vaccine contributes to immunity against malaria [67]. Another approach to prevent malaria infection, using mRNA vaccines, was attempted by Ganley et al., who immunized mice in order to induce liver tissue-resident memory T cells (Trm cells) cells to prevent malaria. More specifically, vaccination of mice with mRNA encoding the model antigen chicken ovalbumin (OVA) and the malaria antigen RPL6, incorporating an active fluorescent derivative of αGC, BODIPY-αGC (αGCB), induced memory T cell response with strong Trm cells production in the liver. Moreover, vaccinated mice upon challenge with *P. berghei* sporozoites that target the liver, demonstrated high level protection from blood-stage infection [68].

Finally, another study identified *Plasmodium falciparum* glutamic acid-rich protein (*Pf*GARP) as a potential candidate that was tested using an mRNA-based vaccination strategy. *Pf*GARP is present on the surface of infected red blood cells and can be identified by antibodies found in children who exhibit relatively higher resistance to *P. falciparum* infection, as they induce programmed cell death in the red blood cells. In a study performed by Raj et al., the effectiveness of the mRNA-LNP *Pf*GARP vaccine was assessed in Aotus monkeys. Three doses of vaccine reduced parasitemia in Aotus monkeys upon challenge with RBCs infected with blood-stage *P. falciparum* FVO strain [69].

Both mRNA and self-amplifying RNA vaccines have the potential to be the next-generation malaria vaccines. Self-amplifying RNA vaccines have been shown to be significantly more immunogenic in comparison with mRNA formulas. Due to this advantage, maybe there would not be the need for multiple booster doses that are more cumbersome to administer, especially outside of routine vaccination schedules in resource-limited areas.

In light of the recent achievements with mRNA-based vaccines targeting SARS-CoV-2, along with the endeavors of numerous researchers who have embraced a similar approach for developing vaccines against malaria, the field is now ripe for pharmaceutical companies to pursue an mRNA strategy in the fight against malaria and other infectious diseases.

## 7. Challenges and Future Perspectives in Malaria Vaccine Development

Malaria vaccine development has been challenging throughout the years primarily due to the complex biology of the parasite and the intricate host immune response required for protection but also due to the evolution of resistant parasites, which escape treatment and vaccine protection. Furthermore, as socioeconomic factors undergo constant changes worldwide and climate change continues, parasitic diseases such as malaria exhibit adaptability and evolution, allowing them to exploit these newly emerged factors to their advantage.

The introduction of the Mosquirix malaria vaccine and the advancements in mRNA vaccine technology have paved the way for a more promising future in the battle against this devastating disease [14]. Although mRNA vaccine technology seems to be the answer to antimicrobial resistance and fast-tracking vaccine developments [70,71], it also has limitations, namely poor stability and difficulties in delivering in areas of interest. Moreover, excessive immunogenicity should be avoided in order to overcome a possible host inflammation or autoimmunity [72].

Advances in nanoparticle-based mRNA delivery and, more specifically, the use of lipid-based nanoparticles have allowed mRNA-based therapeutics to become more clinically viable and relevant as a strategy for hepatotropic infectious diseases such as malaria [72,73]. The self-amplifying mRNA vaccine approach has instilled hope in surpassing the high immunogenicity observed in conventional mRNA vaccines. This method entails administering a small initial dose of mRNA that can be amplified upon insertion in the cells, and it will potentially reduce the necessity for multiple vaccine boosters [6]. Hence, there is an attempt to use the self-amplifying mRNA strategy for the development of a malaria vaccine. It is noteworthy that BioNTech, leveraging its success in creating the SARS-CoV-2 vaccine, has announced the commencement of clinical trials for mRNA vaccines targeting malaria. More specifically, BioNTech company has already developed the BNT165b1 vaccine, a ribonucleic acid (RNA)-lipid nanoparticle (LNP) encoding for part of the *Pf*CSP and Phase I clinical trial for assessing its safety, tolerability, and immunogenicity has already been designed (NCT05581641) [74]. This approach holds promise for the future due to its ability to forgo the need for freezer storage, making it more feasible for delivery and administration in infectious diseases endemic regions [6,72].

Overall, malaria vaccine development efforts have shown accelerated progress in the last few years. The RTS S/AS01 vaccine has been approved, and there are promising candidates, such as R21 and *Pf*SPZ. Development should continue with the aim of generating a malaria vaccine with high efficacy and improved durability of protective response. In this regard, the commitment of necessary resources such as regulatory approval, implementation, and financing are becoming increasingly important for eliminating malaria.

## Figures and Tables

**Figure 1 vaccines-11-01452-f001:**
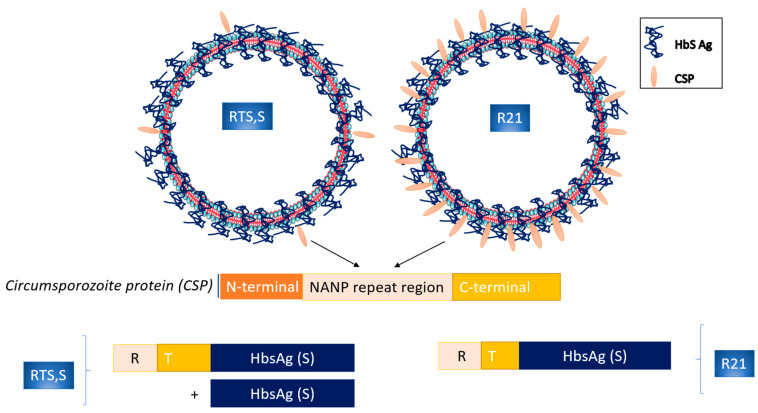
Schematic representation of the circumsporozoite protein subunit vaccines RTS,S and R21 structure.

**Figure 2 vaccines-11-01452-f002:**
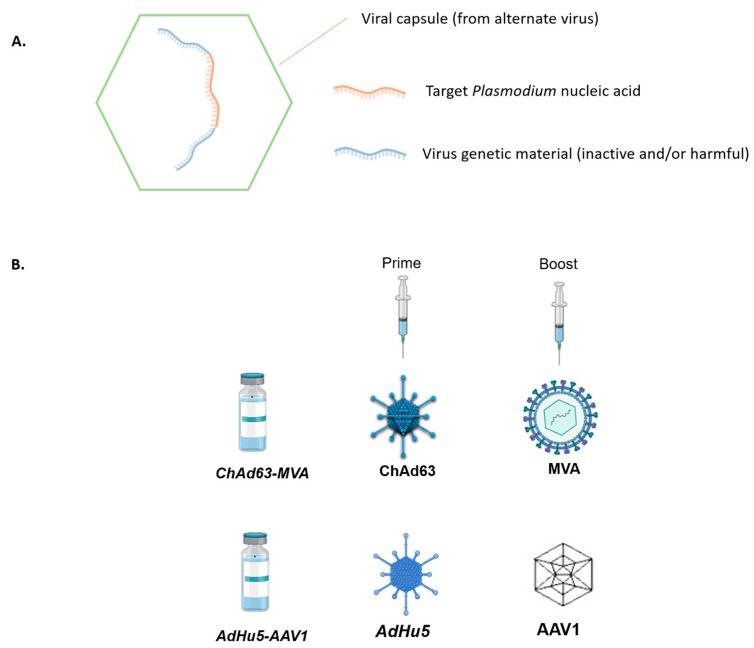
Schematic representation of viral-vector vaccine components. (**A**) Schematic representation of general viral-vector vaccine components. In a viral capsule usually from alternate virus (green), a recombinant DNA is incorporated which contains part of a virus genetic material (inactive and/or harmful) (blue) and the target *Plasmodium* gene. (**B**) Schematic representation of two prominent viral-vector-vaccine approaches against malaria in humans.

**Figure 3 vaccines-11-01452-f003:**
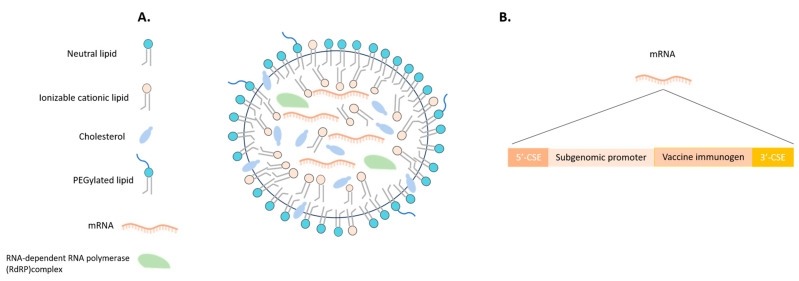
Schematic representation of potential self-amplifying mRNA-LNP vaccine components. (**A**) Schematic representation of an LNP formulation. The membrane consists of neutral lipids, polyethylene glycol lipids (PEGylated lipids), cholesterol and ionizable cationic lipids. For the self-amplifying mRNA-LNP vaccine, in the formulation the target mRNA must be included along with RNA-dependent RNA polymerase (RdRP) complex which will enable the intracellular RNA amplification and the consequently abundant expression of the target protein upon the immunisation of the host. (**B**) The target self-amplifying RNA encodes 5′ and 3′ CSE (conserved sequence element) sequences, a subgenomic promoter and the vaccine immunogen.

**Table 1 vaccines-11-01452-t001:** Major characteristics of the sporozoite vaccine formulations.

Vaccine	Company	Target	Source of Peptide Antigens	Adjuvant	Immune Response	Clinical Status
RTS,S/AS01_Ε_	GlaxoSmithKline Biologicals SA	Inhibition of sporozoite infection	Circumsporozoite protein	AS01	Protective humoral/cellular	WHO recommended
R21	Serum Institute of India (Pune, India)	Inhibition of sporozoite infection	Circumsporozoite protein	Matrix-M	Protective humoral	Phase III clinical trial
*Pf*SPZ	Sanaria Inc.	Killing of infected hepatocytes	Whole sporozoite	None	Protective humoral/cellular	Phase II clinical trial
*Pf*SPZ-CVac (CVac = Chemoprophylaxis Vaccine)	Sanaria Inc.	Killing of infected hepatocytes	Whole sporozoite	None	Protective humoral/cellular	Phase II clinical trial

## Data Availability

Not applicable.

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
