# Peer review of "Malaria Vaccines: From the Past towards the mRNA Vaccine Era"

_vaccines, 2023, doi:10.3390/vaccines11091452_

Round 1

Reviewer 1 Report

The title of the review is misleading.  The review has mainly concentrated on sporozoite vaccines, brushing briefly on genetically attenuated vaccines (I believe this reference has to be included since this reports on a CHMI study: DOI: 10.1126/scitranslmed.abn9709; a viral vectored vaccine was also missed: DOI:10.1126/scitranslmed.aaa2373) prior to reporting on mouse studies. 

To be balanced, the authors should consider efforts from other stage vaccines (e.g. blood-stage vaccines: DOI: 10.3389/fimmu.2022.978591,  DOI: 10.1080/21645515.2015.1026496)  as wewll a CHMI study with a P. vivax vaccine candidate: DOI:10.1126/scitranslmed.adf1782.  Clearly the review scope has to be defined by the authors and the title of the review to be adjusted as most of the works on the mRNA vaccine are still on the early stages, but the title seems to imply that there are already human trials going on. Moreover if the emphasis is on mRNA then it is best to have a discussion on how these are very different (and I think a bit premature compared to RTS/S, R21 and PfSPZ--- following the presentation from the authors. One should care especially if this will be read/interpreted by young researchers).

These mRNA preliminary works are also not cited:

  1. Ganley, M., Holz, L.E., Minnell, J.J. et al. mRNA vaccine against malaria tailored for liver-resident memory T cells. Nat Immunol (2023). https://doi.org/10.1038/s41590-023-01562-6

  1. Chuang, YM., Alameh, MG., Abouneameh, S. et al. A mosquito AgTRIO mRNA vaccine contributes to immunity against malaria. npj Vaccines 8, 88 (2023). https://doi.org/10.1038/s41541-023-00679-x

  1. Nirbhay Kumar et al, mRNA-LNP expressing PfCSP and Pfs25, two leading vaccine candidates targeting infection and transmission of Plasmodium falciparum, npj Vaccines (2022). DOI: 10.21203/rs.3.rs-1895368/v1, http://dx.doi.org/10.1038/s41541-022-00577-8

  1. Ana Maria Valencia-Hernandez et al, A Natural Peptide Antigen within the Plasmodium Ribosomal Protein RPL6 Confers Liver TRM Cell-Mediated Immunity against Malaria in Mice, Cell Host & Microbe (2020). DOI: 10.1016/j.chom.2020.04.010

Reviewer 2 Report

Thank you for the opportunity to review the manuscript entitled "Malaria vaccines: from the past towards the mRNA vaccine era".

Please find below comments and suggestions relevant to this manuscript.

Abstract:

Line 8: Please rephrase as:

Plasmodium spp. are the etiological agent of malaria, a life-threatening parasitic disease transmitted by infected mosquitoes.  Malaria remains a major global health challenge, particularly in endemic regions.

Line 10: Please rephrase as

Over the years, various vaccine candidates targeting different stages of Plasmodium parasite life-cycle have been explored, including subunit vaccines, vectored vaccines, and whole organism vaccines, with Mosquirix, a vaccine based on a recombinant protein, as the only currently approved vaccine for P. falciparum malaria.

Line 22: It explores recent studies and advancements in mRNA-based malaria vaccines, emphasizing their ability to induce robust immune responses and their suitability for delivery in resource-limited, malaria-endemic areas.

Note: Please rephrase as

It explores recent studies and advancements in mRNA-based vaccines for malaria.

Introduction section:

Line 29: Malaria, a life-threatening mosquito-borne infectious disease caused by the Plasmodium parasite, continues to pose a significant global health burden as half of the world’ population

……half of the world’s population,,,

Line 33: Four different parasite species cause malaria….

Five plasmodium species can cause malaria in humans, namely…..

Line 39: …..80% of all malaria deaths were accounted to children under 5 years old in the African continent

Rewrite ….80% of all malaria deaths were accounted in children under 5 years old in the African continent

Line 46: Remove ….What is more..

Line 49: Subsequently, the sporozoites invade the hepatocytes in the liver where they mature and replicate up to the rupture of the hepatocyte and the release of the merozoite form which eventually invade in the erythrocytes to transform from rings, trophozoites to schizonts and back to merozoites that can infect more blood cells.

Please rephrase this statement as:

Subsequently, the sporozoites invade hepatocytes in the liver where they mature, replicate and release  merozoite forms which eventually invade erythrocytes where they transform  from rings, trophozoites to schizonts stage.  The ruptured schizonts release merozoites that can infect fresh blood cells.

Line 55: Global Technical Strategy for Malaria (use capital letters).

Line 56: Although, vaccination represents a crucial pillar in the comprehensive approach to malaria control, the development of an effective malaria vaccine has been an elusive goal, primarily due to the complex biology of the parasite and the intricate host immune response required for protection.

Suggested…..development of an effective malaria vaccine has been difficult,…..

Line 59: Re-write as

However, recent advancements in vaccine research and the identification of promising vaccine candidates have renewed hopes for developing effective vaccines for malaria.  

Line 63:  Remove this line: ‘By synthesizing existing knowledge and incorporating recent scientific advancements’,

…this review will contribute to a deeper understanding of the challenges posed by malaria and the potential solutions that can be harnessed for effective vaccination development

Rephrase: …this review aims to discuss the challenges posed by malaria and new strategies that can be used to develop effective malaria vaccines.

Line 77: Vaccines that target sporozoites aim to prevent or impair infection of the liver in order to induce immunity against infection and disease

Rephrase as : Vaccines that target sporozoite stage aim to prevent infection of the liver and induce immunity against the parasites.

Line 78: Please check the language. Whole killed sporozoites were one of the first malaria immunogens tested as a malaria vaccine [8]. These early efforts targeted the pre-erythrocyte stage of the complex life cycle of Plasmodium parasites  and offered only partial protection exposing the vaccinated animals susceptible to blood stage challenge [8].

Also, original references should be cited for fundamental statements like the one above (instead of referring to a review).  

The radiation-attenuated sporozoite approach was rapidly translated to humans in a series of dose and regimen optimization studies [8]. Please use correct reference.

Line 87: P. falciparum (Pf) sporozoite vaccine (PfSPZ) from the NF54 isolate manufactured by Sanaria Inc. (Rockville, MD, USA) met all the regulatory requirements for safety and consistency regarding human use.

Rephrase as: P. falciparum (Pf) sporozoite vaccine (PfSPZ) from the NF54 isolate manufactured by Sanaria Inc. (Rockville, MD, USA) met all the regulatory requirements needed for testing in humans.

Line 89: Although animal models do not adequately reflect malaria, studies in animals revealed that PfSPZ should be administered only intravenously (IV) in order to confer protection.

Rewrite : Studies in animals revealed that PfSPZ should be administered only intravenously (IV) in order to confer protection.

Lines 92-94: Please remove. ‘The implementation of this model especially in malaria-endemic settings has many advantages towards the natural parasite exposure since it permits the deeper comprehension of the interplay between vaccine-induced immunity and pre-existing immunity’.

Line 97: Consequently, studies in malaria-naïve volunteers who received IV doses of PfSPZ vaccine …..

Rrewrite as: Consequently, studies in malaria-naïve volunteers in the US who received four doses of PfSPZ vaccine provided evidence for long-term protection (up to 14 months) by inducing sterile…..

Line 122: Please use original reference.

Line 125: Another vaccine approach combined replication–intact sporozoites with anti- malarial drug prophylaxis such as chloroquine and mefloquine to prevent malarial illness [8].

Please cite the original published paper.  

Line 130: Use correct reference for :

PfSPZ-CVac is a stronger immunoantigen compared to PfSPZ since the infectious PfSPZ in PfSPZ-CVac expresses 4500 different proteins, including blood stage proteins whereas PfSPZ expresses ~1000 proteins. Thus there are dramatically more parasites and antigens presented to the immune 133 system per PfSPZ injected with PfSPZ-CVac [8].

Also, please clarify for the reader the difference between PfSPZ (replication-deficient) and PfSPZ-CVac where the sporozoite is infectious but is controlled by a malaria drug.  

Line 149: Write as ‘Another P. falciparum genetically….’

Line 151:  Pf double knockout (PfΔb9Δslarp) SPZ was manufactured by Sanaria (PfSPZ-GA1 Vaccine) and first clinical trials showed that it can be safely administered to malaria-naïve volunteers by DVI [21].

Please mention results in brief for this study.

Circumsporozoite protein subunit vaccines section:

RTS,S/AS01E written as RTS,S/AS01E  

Line 166-169: Only Ref 25 for this good.

Line 185: Write as… ‘….among malaria vaccine recipients over 12 months…’

Line 201: Write as  ‘…..protection as determined by CHMI in malaria-naïve individuals.’

Line 205-210: Please rephrase this to avoid contradictory statements for the same vaccine, such as..significant success, lack of efficient and durable response etc. Also, cite relevant original references.

Line 213-215: Write as  ‘…..fast-track strategies in vaccine research ….’

Also, please rephrase

Line 217-220: This statement does not appear to fit into the context.

Line 224: Please delete ‘as targets of protective T-cell response’ from the sentence.

Line 224-228: Please rephrase appropriately.

Line 234: Remove ‘simultanously’

Line 254: Remove ‘Hence’

Malaria vaccines in mRNA era section:

Line 270: The perceived barriers of mRNA vaccines, including the challenges stemming from technological advancements in RNA biology and chemistry, are now being overcome, allowing for the widespread implementation of this technology.

Please rephrase to have a correct interpretation of this statement.

Line 227: remove ….’alongside the antigen’.

Line 279: remove  ….. ‘with initial low dosages of the antigen for maximum immunization and minimization of implications’

Line 285:…..to administer…..

change to  …..to develop…..

Line 286: Thus, leveraging the recent achievements in vaccine development for COVID-19, RNA technology is now being utilized to administer hopeful antimalarial candidate vaccines. For instance, a vaccination strategy is being built on the antigen Plasmodium macro phage migration inhibitory factor (PMIF) which is secreted by the parasite and serves to suppress the host's inflammatory response to the infection, particularly targeting the T-cell response [47].

Reference number 47 is from 2018, but its work is described as an example coming after the success of COVID-19 RNA based vaccine. Please check. Ref No 48 is better suited as an example in this context.

Line 304:  This remarkable outcome positions it as an exceptionally promising candidate for a malaria vaccine.

Rephrase: This remarkable outcome positions it as a promising candidate for a pre-erytrhocytic malaria vaccine.

Line 310: In fact, the administration of Pfs25 mRNA-LNPs resulted in IgG production that effectively prevented malaria transmission to mosquitoes. Additionally, after undergoing four immunizations with PfCSP mRNA-LNP, the mice demonstrated protection against sporozoite challenge.

Rewrite: Pfs25 mRNA-LNPs induced antibody responses that effectively prevented malaria transmission to mosquitoes and four immunizations with PfCSP mRNA-LNP protected mice against sporozoite challenge.

Line 328-329: Plasmodium falciparum (Italics)

Write as   ‘….as a potential candidate that was tested using mRNA based vaccination strategy.’

Line 329: PfGARP is present on the surface of infected red blood cells and can be identified by antibodies found in children who exhibit a relatively higher resistance to P. falciparum infection, as they induce programmed cell death in the red blood cells.

Rephrase as: PfGARP is expressed on P. falciparum-infected red blood cells and can be identified by antibodies found in children who exhibit a relatively higher resistance to P. falciparum infection.

Line 332: Acquits Therapeutics assessed the effectiveness of the mRNA-LNP PfGARP vaccine in Aotus monkeys and demonstrated that it reduced the parasitaemia upon a 3-dose regimen of 50 μg.

This statement does is not correct.

Reference No 55 does not mention Acuitas assessed the effectiveness of mRNA-LNP PfGARP vaccine in Aotus monkeys. The paper mentions their role in ‘mRNA-based vaccine design and production’. Please check the paper.

Further, …..demonstrated that it reduced the parasitaemia upon a 3-dose regimen of 50 μg

Correct as: Three doses of vaccine reduced parasitaemia in Aotus monkeys upon challenge with RBCs infected with blood stage P. falciparum FVO strain.  

Line 348: The introduction of the Mosquirix malaria vaccine and the advancements in mRNA vaccine technology have paved the way for a more promising future in the battle against this devastating disease although now the need for more effective vaccines is more pressing than ever.

Amend as: The introduction of the Mosquirix malaria vaccine and the advancements in mRNA vaccine technology have paved the way for a more promising future in the battle against this devastating disease.

Line 353:  Remove….’excessive immunogenicity’

Line 361: Hence, the majority of researchers and companies are concentrating their efforts on the self-amplifying mRNA strategy for the development of a malaria vaccine.

This statement does not appear correct.

May rephrase as: Hence, there is attempt towards using mRNA strategy for the development of a malaria vaccine. It is noteworthy that BioNTech, leveraging its success in creating the SARS-Cov-2 vaccine, has announced the commencement of clinical trials for mRNA vaccines targeting malaria.

Line 365:-367: Remove ‘This approach holds promise for the future due to its ability to forgo the need for freezer storage, making it more feasible for delivery and administration in malaria-endemic regions [6,58]’

Note: While there is attempt towards making stable mRNA based vaccines that do not require ultra low temperatures, there is no mention of such a statement in the literature for malaria vaccines.

--------------------------------------

Other comments/suggestions

Original references should be cited when referring to specific findings and facts. For example, reference no 8 (a review article) is used a number of times and in extended portions in the manuscript to refer to major results from previous studies.

In some portions of the manuscript, the overall English language used needs to be concise and corrected at places, and repeat statements with the same information or interpretation should be avoided.

Reviewer 3 Report

The manuscript by Tsoumani et al. reviews studies aimed at the use of mRNA vaccines in malaria prevention along traditional vaccine formulations and constructs. The review is timely and highlights the need for more studies aimed at identifying vaccine candidates that can be developed using the mRNA platform.

Comments:

1.      The review does not clarify the current status of the major vaccines such as the different sporozoite vaccine formulations. Additional detail from the literature is needed to document the progress (or lack) and limitations of the sporozoite vaccines. Provide a table for the sporozoite vaccines and the current status for each type. What are the current mRNA vaccines for malaria in the pipeline? 

2.      The major similarities and differences between RTS,S and R21 vaccines need further explanation. Provide a figure showing vaccine structural components for both vaccines.

3.      For viral-vector vaccines and self-amplifying RNA vaccines, figures are also needed to clarify the components of the constructs.

Line 12: “consisting” instead of “consisted”

Line 21: Are there current limitations for the use of mRNA vaccines in “resource-limited malaria endemic areas”?

Line 29: “world’s” population

Line 33: Revise to five human Plasmodium species. The zoonotic P. knowlesi is the fifth human Plasmodium species.

Lines 55-56: several “strategies”

Lines 67-68: Revise “…in the combat against malaria…”

Lines 76-77: “…different stages of Plasmodium species”

Lines 179-181: Revise this section for clarity. Grammar and punctuation revision will help with clarity.

Lines 182-192: Provide more information regarding the structure and components of the R21 vaccine and its current approvals and use in endemic areas.

Lines 193-202: Provide more information regarding viral-vector vaccines. Explain the types of constructs prepared, trials performed and the levels of success, efficacy, etc.

Line 216: “…30 years ago in (the) basic research.” Delete “the”.

Line 219: “…to (a) newly or resistant emerging pathogens.” Delete “a”.

Lines 220-222: Provide additional explanation for the approach used for DNA vaccines, the typical source of Plasmodium DNA and the process of generating and validating constructs. What is the current status of DNA vaccines for malaria? This should be included in the review.

Line 227: Provide an explanation for “potential implications”. This is currently vague.

Lines 248-249: Revise “…depending on the method”

Lines 267-295: Specifically, how has the self-amplifying RNA strategy been used in malaria vaccine studies? Is PMIF the only target antigen investigated using self-amplifying RNA? How is self-amplifying RNA different from mRNA-LNP vaccines?

Lines 279-280: Provide an explanation for “…minimization of implications”. This is currently vague.

Lines 346-347: Provide an explanation for “…newly emerged factors” What are examples of these factors?

Lines 353-354: What are some of the adverse implications of “excessive immunogenicity”? In particular, how does this develop with the use of mRNA vaccine platforms?

Lines 62-73: Section 6-Strenghten the future perspectives section to include suggestions, or implications of the current status of vaccine studies for policymakers, healthcare professionals, public health officials, etc. Is there realistic progress in malaria vaccine studies? Are we any closer to eliminating malaria?

Minor editing required.

Round 2

Reviewer 1 Report

The paper is well revised.

Reviewer 3 Report

The authors have responded to the previous queries.

Minor comments and questions:

Line 45: Plasmodium

Line 67: Targeting

Line 68: "Vaccines that target the sporozoite stage..."

Lines 69-70: Check spaces in the sentence.

Lines 93-96: Rephrase this section to clarify effectiveness of PfSPV in infants.

Lines 137-138: Rephrase to clarify study cited in ref. 26

Lines 186-187: Rephrase to clarify R21/MM vaccine data in African children.

Line 207: "...last years the progress..." Is this referring to a specific year, e. g. 2022, or last several years? Please clarify.

Line 236: "...as quickly as..."

Line 241: "...as well as highly immunogenic..."

Line 246:  "...and nowadays more than one..."consider changing to "currently"

Line 257: delete "."

LIne 262: delete "has"

Lines 290-291: insert a space between "response" and "Wang".

Line 346: Delete "the" before "P. falciparum".

Line 439: delete "an" and replace with "a ribonucleic acid..."

Line 444: "...accelerated progress the last years." Please clarify what is meant here.

Lines 521-522: A title is missing from the reference.

The manuscript should be checked for grammar, syntax and use of prepositions before resubmission.
